YOLOv8-POS: a lightweight model for coal-rock image recognition

Zhao Yanqin zyq_jean@usth.edu.cn
http://orcid.org/0009-0000-0713-0738 Wang Wenyu
School of Computer and Information Engineering, Heilongjiang University of Science and Technology , Harbin, Heilongjiang , China
Fontana Simone
Electronic publication date: 2025 Apr 7
Publication date: 2025
Volume: 11
Electronic Location ID: e2820
Received 2024 Oct 23; Accepted 2025 Mar 20
Copyright: © 2025 Zhao and Wang
Copyright year: 2025
Copyright holder: Zhao and Wang
License: This is an open access article distributed under the terms of the Creative Commons Attribution License, which permits unrestricted use, distribution, reproduction and adaptation in any medium and for any purpose provided that it is properly attributed. For attribution, the original author(s), title, publication source (PeerJ Computer Science) and either DOI or URL of the article must be cited.
License URL: https://creativecommons.org/licenses/by/4.0/

Keywords: Attention mechanisms, Coal-rock image recognition, Deep learning, Object detection, YOLOv8

Funding: Basic Research Operating Costs of Undergraduate Colleges and Universities in Heilongjiang 2022-KYYWF-0565 Heilongjiang University of Science and Technology 2024 College Students’ Innovation and Entrepreneurship Training Program Project This work was supported by the Basic Research Operating Costs of Undergraduate Colleges and Universities in Heilongjiang Province Project (No.2022-KYYWF-0565) and Heilongjiang University of Science and Technology 2024 College Students’ Innovation and Entrepreneurship Training Program Project. The funders had no role in study design, data collection and analysis, decision to publish, or preparation of the manuscript.

==============================
A novel approach, designated YOLOv8-POS, is introduced to address the issue of false detections in coal-rock image recognition tasks, frequently caused by factors such as image defocus, dim lighting, and worker occlusion, and to further enhance the model’s accuracy and reduce its complexity. The methodology introduces a C2f-PConv module, which ingeniously combines the strengths of C2f and partial convolution (PConv) to selectively process channels. This reduces unnecessary computational overhead while preserving the integrity of critical feature information, thus significantly cutting down on the model’s parameters and computational demands. Additionally, an Overlapping Spatial Reduction Attention module is incorporated into the model’s architecture to optimize the fusion of spatial features, substantially improving the handling of complex scenarios. The adoption of a slim-neck design further streamlines the computational and storage requirements, leveraging meticulously engineered lightweight modules to enhance the model’s practical applicability. Empirical results demonstrate that YOLOv8-POS markedly improves performance on coal-rock image datasets, achieving an AP50 of 77.1% and an AP50:95 of 63.6%, while concurrently reducing the model’s parameters to 2.60 M and the floating point operations (FLOPS) to 6.4 G. Comparative evaluations with other prominent algorithms confirm the superior performance of this refined approach, solidifying its advantage in practical deployments.

Introduction

The efficiency of surveying and mining coal has a direct impact on the stability of energy supply and economic benefits, as coal is a crucial component of global energy consumption (Longwell, Rubin & Wilson, 1995; Dai & Finkelman, 2018). Within this specific framework, the utilization of coal and rock image recognition technology is crucial to fulfill the requirements for high precision, low latency, and minimal resource consumption in coal-rock recognition within mining regions. This technology effectively discerns the attributes of coal and rock, significantly enhancing the speed of recognizing and processing coal resources, while minimizing the requirement for manual intervention, lowering resource depletion, and mitigating environmental contamination (Finkelman, Wolfe & Hendryx, 2021). Furthermore, the utilization of coal and rock image recognition technology can enhance the whole mining process, enhance the precision and productivity of coal mining, and contribute to the attainment of environmental conservation and sustainable development objectives (Xie et al., 2021).

However, in practical applications, due to the diversity of mining scenarios and the limitation of equipment settings, the recognition of coal-rock images meets many problems, which adversely affect the recognition accuracy and efficiency. Common issues encompass image blurring, low light conditions, obstruction by workers, and submersion in water with coal-rocks. Image defocus and low light conditions can lead to the distortion of texture and structural information in coal-rock, thereby affecting the accuracy of classification and recognition. The presence of individuals or equipment obstructing the view complicates image processing and reduces the reliability of the recognition system. Additionally, coal-rocks often become water-saturated during mining, leading to changes in surface colour and increasing the complexity of image processing. Figure 1 presents several image examples. Such situations may result in reduced recognition accuracy and delayed real-time responses, thereby limiting the effective utilization and safe extraction of coal resources.

Figure 1 (A–D) Demonstration of the difficulties of coal-rock image recognition.

Image credit: Wang et al. (2024). Coal and Rock [Data set]. Zenodo. https://doi.org/10.5281/zenodo.10702704.

While existing literature has sought to enhance coal-rock image recognition, the majority of these efforts concentrate on isolated issues and do not comprehensively address the dual demands of minimal resource consumption and optimal real-time performance in the practical context of mining environments. This study addresses these issues by implementing the YOLO (You Only Look Once) algorithm for target detection, specifically utilizing the YOLOv8 version (Ultralytics, 2023), which markedly improves the processing of intricate image scenes through advancements in network architecture, training strategy, and optimization algorithm. Based on the efficiency and accuracy of YOLOv8, this study proposes a lightweight coal-rock image recognition method named YOLOv8-POS. The method retains the benefits of YOLOv8 while incorporating an optimization strategy for coal-rock images, thereby enhancing performance and practical applicability. In comparison to the original YOLOv8 model, YOLOv8-POS demonstrates notable performance enhancements on the coal-rock image dataset, with improvements of 2.4% in AP50 and 4.1% in AP50:95. Concurrently, the model’s parameter count and FLOPs are reduced by 0.41 M and 1.8 G, respectively, thereby substantiating the efficacy of the current approach in terms of both accuracy and processing speed. This article’s primary contributions are as follows: (1) The proposed C2f-PConv module incorporates partial convolution (PConv) into the C2f module. By effectively utilizing the redundancy of the feature maps, it successfully reduces the model’s cost without compromising its accuracy. This enables efficient and accurate recognition of complex coal-rock images.

(2) The Overlapping Spatial Reduction Attention (OSRA) is added, which recognizes and highlights key parts of the features so that the model focuses on the key features in the image, and also significantly strengthens the network’s ability to process high-resolution input images through its overlapping design, enhancing the model’s ability to locally detail and globally contextually integration ability, thus better capturing the complex textures in coal-rock images.

(3) The slim-neck paradigm is introduced as a method to improve the integration of features and the flow of information in the model by replacing many modules. This approach efficiently reduces the computational and parametric aspects of the model, making it better suited for downhole equipment with limited resources.

The rest of the article is organized as fo9llows: The “Related Work” section examines research pertinent to coal-rock image recognition. The “Materials and Methods” section explains the principles of the enhanced YOLOv8-POS model. The “Experimental setup” section describes the preparation of the experiment. The section “Results and Discussion” gives the corresponding experimental results and discussion. Lastly, the “Conclusions” section encapsulates the entire work and anticipates future developments.

Related work

This chapter reviews conventional and machine learning methods for coal-rock image recognition, then examines deep learning-based methods like convolutional neural networks, two-stage, and one-stage object detection algorithms to demonstrate their efficacy.

Traditional methods and machine learning

Prior to the widespread adoption of deep learning techniques in image recognition, the identification of coal-rock images primarily relied on conventional methods, which involved manual extraction of image features and the application of fundamental machine learning algorithms. Conventional methods generally require specialised knowledge to select and manipulate image characteristics for diverse identification purposes.

Sun & Su (2013) extracted 22 coal-rock texture features using a grey scale co-occurrence matrix. Using feature selection, these features were reduced to four important features. Fisher’s discriminant method categorised coal-rock images with 94.12% accuracy. This strategy greatly improved sample distinction. Sun & Chen (2015) employed a wavelet domain-based asymmetric generalised Gaussian model together with an enhanced relative entropy similarity measure to categorise coal-rock images in their study. They achieved an average recognition rate of 87.77%. Nevertheless, this approach is highly susceptible to interference from noise. Wu & Tian (2016) employed a dictionary learning algorithm and KNN classification algorithm to extract features and classify coal-rock images. The dictionary was initialised and updated through random selection, effectively representing the features of coal-rock images. However, this approach is computationally expensive and lacks strong generalisation ability. Wang & Zhang (2020) utilised local binary pattern (LBP) and grey level covariance matrix (GLCM) methods to analyse texture variations among various coal-rock types. Four texture features were identified: energy, entropy value, contrast, and inverse differential moment, which effectively detected differences in coal-rock characteristics. Further enhancements are necessary to improve the robustness of the analysis. Zhang et al. (2022) utilised Gaussian filtering on gangue greyscale images and employed least squares vector machines to create recognition models by integrating various attributes. The results demonstrate that the gangue image achieves an identification rate of 92.2% and 91.5% when utilising grey scale skewness and texture contrast as markers, respectively.

While these initial methods showed potential, manual feature extraction techniques rely primarily on the designer’s expertise and preferences, and have limitations in terms of adaptability and scalability. Moreover, when dealing with intricate visual data, conventional machine learning algorithms often encounter challenges related to overfitting and parameter adjustment.

Convolutional neural network

Deep learning techniques have advanced the use of convolutional neural networks (CNNs) in image recognition. CNNs play a major role in improving the accuracy and speed of recognition by automating the process of feature extraction.

Huiling & Xin (2019) employed wavelet transform and BP neural network to classify coal-rock images, resulting in a recognition rate of 96.67%. This was accomplished by configuring the number of hidden layer nodes to 10 and performing 500 iterations. Liu et al. (2019) utilised a combination of a multi-scale completed local binary pattern (CLBP) and a CNN-based deep feature extraction (DFE) technique to recognise coal-rock images. They achieved a recognition accuracy of 97.9167% by employing the cardinality distance of the nearest-neighbour classifier. Pu et al. (2019) attained an accuracy of 82.5% in the recognitionof coal and gangue images by employing a CNN-based VGG16 network in conjunction with migration learning approaches. In their CNN model for coal-rock recognition, Si et al. (2020) incorporated dropout, weight regularisation, and batch normalisation approaches, which led to an enhanced model F1 score of 78.62%.

These studies provide evidence of the successful implementation and impressive outcomes of convolutional neural networks in the domain of coal-rock image recognition. The researchers achieved a substantial improvement in recognition accuracy and model performance by using advanced feature extraction methods and optimised network designs.

Two-stage object detection

A single classification method no longer meets the need for effective detection and localisation in complex application contexts due to technological advances and rising expectations. This speeds up object detection technology, which recognizes and accurately positions objects in an image. Here, two-stage detection methods such Faster R-CNN (Ren et al., 2016) and Mask R-CNN (He et al., 2018) have been developed. Classification and bounding-box regression are performed on regions of interest (RoIs) created by these approaches. Object detection accuracy improves greatly with this method.

Hua, Xing & Zhao (2019) were able to attain an 88% mean average precision (mAP) for recognising coal seam boundaries using the Faster R-CNN algorithm with the VGG16 network. Shan et al. (2022) added CBAM to ResNet50 to improve Faster R-CNN. This improvement improves the accuracy of recognizing mixed and releasing coal-gangue masses in fully mechanised caving. This improvement raised the model’s average detection and recall rates to 82.63% and 86.53%. In addition, the F1-score rose 7%. Cao et al. (2024) enhanced the Mask R-CNN for the purpose of segmenting coal and gangue. They achieved this by introducing a multichannel Forward-Linked Confusion Convolution Module (MFCCM) and a multiscale high-resolution feature pyramid network structure. Additionally, they presented a multiscale mask head structure. The enhanced algorithm achieves a precision of 97.38%, exhibiting a 1.66% increase in comparison to the initial model.

Despite the great accuracy offered by two-stage detection approaches in object detection, their computing demands and poor processing speed still hinder their widespread use in real-time applications. These strategies must be further improved to decrease latency while preserving accuracy for real-time processing.

One-stage object detection

One-stage detection methods, including single shot multibox detector (SSD) (Liu et al., 2016) and YOLO (Redmon et al., 2016), have been widely examined in object detection for their high efficiency and low computational demands. These technologies significantly improve operational efficiency and processing speed by accurately recognizing and locating the bounding boxes and categories of items within images. Technological advancements and iterative algorithm updates have led to various versions of the YOLO model demonstrating outstanding performance in coal-rock image detection.

Zhang et al. (2020) employed the YOLOv2 algorithm to recognize images of coal-rock in downhole conditions. They achieved an identification accuracy of 78% and a detection speed of 63 frames per second, surpassing the performance of Faster R-CNN and SSD. Sun et al. (2022) enhanced the YOLOv3 algorithm by incorporating depth-separable convolution and cubic spline interpolation algorithms. This resulted in a 5.85% improvement in model accuracy in the x direction and a 16.99% improvement in the y direction. Additionally, the number of parameters was reduced by approximately 80%. Li et al. (2022) improved YOLOv3’s efficacy in coal gangue recognition through the integration of deformable convolution. K-means clustering was employed to improve the accuracy of anchor frame localisation. The optimised algorithm attained a mAP of 99.45% while decreasing the number of FLOPs by 61.4%. Liu et al. (2021) enhanced YOLOv4 for the recognition of coal and gangue by optimising anchor values through cluster analysis and increasing the number of feature pyramid layers, resulting in a 0.81% improvement in the mAP of the modified network. Wang et al. (2022) enhanced YOLOv5 for coal-rock image recognition by incorporating the CBAM attention mechanism and Transformer, achieving a mAP of 92.8%. However, the use of CBAM increased computational complexity and excessively depended on global information, frequently neglecting significant local features. Zhao & Deng (2024) enhanced YOLOv7 for coal-rock image recognition by integrating the ConvNext module with 7 × 7 convolutional kernels, and by incorporating the SimAM attention mechanism and the αIoU loss function, resulting in a 3.9% improvement in accuracy and a 1.5% increase in mAP. These changes improve model performance, but the larger convolutional kernel is computationally expensive and cannot capture delicate features.

In conclusion, one-stage detection models have improved coal-rock image recognition, but they struggle with complicated situations, notably detail capture and feature extraction. Therefore, this study uses and enhances the newest YOLOv8 model to address the accuracy and efficiency issues of existing methods. It uses innovative methods to increase the model’s performance in complicated coal-rock image recognition and classification.

Materials and Methods

This section provides a quick introduction to the YOLOv8 algorithm, followed by the presentation of the revised YOLOv8-POS method. A series of structural improvements will be introduced to enhance the accuracy and efficiency of the model in recognising coal-rock images.

YOLOv8 algorithm overview

Ultralytics has unveiled the YOLOv8 algorithm, which has five distinct models: YOLOv8n, YOLOv8s, YOLOv8m, YOLOv8l, and YOLOv8x. These models vary in terms of network depth and feature map size. This study selects YOLOv8n as the benchmark model due to its superior processing speed. The model is then enhanced to attain both high detection accuracy and a lightweight structure.

The YOLOv8 network comprises four primary components: Input, Backbone, Neck, and Head. The input employs a mosaic data augmentation technique, which enriches the diversity of data by randomly rescaling and merging four images. This improves the model’s capacity to adapt to complicated backdrops with varying scales. The backbone incorporates the faster implementation of CSP Bottleneck with two convolutions (C2f) and spatial pyramid pooling-fast (SPPF) modules. The C2f module draws inspiration from the edge-enhanced local attention network (ELAN) architecture in YOLOv7. It enhances the processing of complicated features by incorporating more branches and cross-layer connections, hence transmitting more comprehensive gradient information. The SPPF module enhances the efficiency of feature extraction by utilising a rapid pooling technique, enabling the processing of input images of any size. The neck employs the C2f module-based path aggregation network with feature pyramid network (PAFPN) architecture. This architecture integrates the top-down feature fusion of feature pyramid networks (FPN) with the bottom-up feature augmentation process of path aggregation network (PAN). This allows for the incorporation of high-level semantic information with low-level detailed information, leading to more efficient feature extraction. The head employs an anchor free strategy and decoupled-head. The anchor free method directly predicts the centroid of the object, eliminating the need for pre-defined anchor boxes and improving the model’s capacity to generalise to targets of different sizes and proportions. The Decoupled-Head algorithm divides the jobs of classification and regression. It uses binary cross-entropy (BCE) Loss for the classification task and complete intersection over union (CIOU) loss with distribution focal loss (DFL) for the regression task.

YOLOv8-POS algorithm

Despite its robust performance across various applications, YOLOv8 exhibits suboptimal results in coal-rock image recognition. This underperformance is largely attributable to inherent challenges presented by the visual similarity of coal and rock images, compounded by the extreme complexity and variability of lighting conditions in underground environments. These factors significantly impede traditional models’ recognition capabilities in such settings. Furthermore, the limited computing resources typically available on terminal equipment in deep coal mines pose substantial constraints on the operational efficiency of complex models. In response, this study introduces specific enhancements to the YOLOv8 algorithm, culminating in the development of the YOLOv8-POS model, which is optimized for coal-rock image recognition. These modifications not only improve recognition accuracy but also contribute to the model’s lightweighting. Figure 2 illustrates the architecture of the YOLOv8-POS model.

Figure 2 YOLOv8-POS network structure.

The C2f-PConv module (Chen et al., 2023) was integrated into the backbone network to minimize unnecessary computations by selectively processing channels, thus preserving vital feature information essential for analyzing coal-rock images, which are characterized by similar textures and intricate details. This module is selected for its capacity to substantially decrease processing demands while effectively maintaining essential image information, thereby striking an optimal balance between precision and computational economy. Concurrently, the OSRA module (Lou et al., 2023) enhances feature fusion in the spatial dimension, significantly improving the model’s ability to discern light-sensitive regions and image edges, thereby boosting adaptability and recognition accuracy in the challenging environments of downhole settings. The OSRA module is introduced as it enhances feature representation while its low computational complexity enables the network to sustain high accuracy without substantially augmenting the computational load. Additionally, the slim-neck paradigm (Li et al., 2024) is incorporated into the feature fusion network, enhancing information flow and feature integration via strategic implementation of the Grouped Shuffle Convolution (GSConv) module and the Variety of View-Grouped Shuffle Cross Stage Partial (VoV-GSCSP) module. This design not only enhances information processing and feature integration but also considerably reduces the computational and storage demands of the model, facilitating more efficient operation on resource-constrained downhole equipment. The adoption of slim-neck is motivated by its flexible and efficient structural design, which reduces model parameters and operational requirements while preserving high detection accuracy, thereby optimizing both accuracy and efficiency. The YOLOv8-POS model significantly enhances coal-rock image recognition performance by integrating the features of each module through targeted enhancements and integration strategies. This sophisticated integration not only boosts the model’s processing capabilities but also achieves further network lightweighting, providing an efficient and precise solution for coal-rock image recognition challenges.

C2f-PConv module

This study enhances the YOLOv8 model by incorporating the PConv module, aimed at reducing the computational expenses and increasing the processing speed of the coal-rock image recognition model. In deep learning architectures, feature maps across various channels often exhibit significant redundancy. Exploiting this redundancy through PConv facilitates improved cost optimization. Unlike Standard Convolution, which employs filters across all input channels thereby increasing both parameter count and computational load, PConv selectively processes certain channels, omitting others. This approach not only improves processing efficiency but also strikes a balance between the computational demands and accuracy, with its parametric and arithmetic requirements positioned between those of Standard Convolution and Depthwise/Group Convolution (Zhang et al., 2023; Qu et al., 2024). The advantages and operational distinctions of PConv compared to other convolution methods are illustrated in Fig. 3.

Figure 3 (A–D) Principles of standard convolution, Depthwise/Group convolution and partial convolution.

Assuming the input and output feature maps possess an identical number of channels, for the input feature map I∈Rc×h×w (where c denotes the number of channels, and h and w signify the height and width of the feature map, respectively), the FLOPs for a Standard Convolution with a kernel size of k×k are presented in Eq. (1):

(1) h×w×k2×c2

Conversely, PConv calculates the initial or last sequential Cp channel as a sample of the complete feature map, with its FLOPs delineated in Eq. (2):

(2) h×w×k2×cp2

In a typical scenario, with the partial ratio r=cpc=14, the FLOPs of PConv constitute merely 116 of those of regular convolution, as demonstrated in Eqs. (1) and (2), resulting in a significant reduction in computing cost.

The PConvBottleneck structure is intended to supplant the traditional Bottleneck in the C2f module, resulting in the enhanced C2f-PConv module. The configuration comprises two PConv modules arranged in series and one shortcut connection. The initial PConv module serves to increase the number of channels, however the subsequent module diminishes the number of channels to align with the output of the shortcut connection situated between the two. The C2f-PConv module preserves the core advantages of the original PConv, notably enhancing spatial feature processing efficiency by reducing redundant computations and memory accesses. This improvement facilitates increased operational speed on edge devices while maintaining high accuracy across various visual recognition tasks, as depicted in Fig. 4.

Figure 4 (A–D) PConv module, PConvBottleneck and C2f-PConv.

The incorporation of the C2f-PConv module within the backbone network markedly decreases the resource expenditure of the model in processing coal-rock images, while preserving exceptional image recognition precision. The module utilises the distinctive benefits of PConv to markedly diminish computational resource demands and enhance processing speed, a technological advancement essential for real-time coal mining applications, allowing edge devices to swiftly and precisely execute intricate coal-rock image recognition tasks, thereby establishing a robust technological foundation for the coal mining sector.

OSRA module

To improve the precision of coal-rock image recognition, OSRA is introduced as an enhancement module. This attention mechanism is enhanced from spatial reduction attention (SRA) (Wang et al., 2021), with its structure illustrated in Fig. 5.

Figure 5 Structure of the OSRA module.

The SRA module decreases computational and storage expenses by decreasing the number of blocks in high-resolution feature maps, with these blocks generally representing a minor segment of the image or feature map. Nonetheless, employing this non-overlapping method for spatial reduction may compromise the spatial integrity around the image boundaries and impact the overall quality of the features. OSRA was proposed to resolve this issue, it employs an overlapping block approach to more effectively preserve and convey spatial information in edge regions. This overlap not only minimises resource consumption but also guarantees that characteristics at the periphery remain undistorted due to inadequate contextual information (Pan et al., 2024; Yao et al., 2024).

Regarding technical implementation, OSRA executes overlapping block processing via depth-separable convolution, maintaining a step size consistent with that of SRA, while the convolution kernel size is deliberately configured to the step size plus three to enhance coverage, overlapping effect, and optimise overall feature quality. Furthermore, OSRA incorporates a local refinement module and a relative position bias matrix, aimed at optimising the attention mechanism to enhance the model’s ability to effectively capture and leverage both local and global information, thereby augmenting the accuracy and efficiency of coal-rock image recognition.

Firstly, the input data X is initially processed using the OSR module to produce new feature data Y. This procedure is termed spatial overlap processing, which preserves edge information by overlapping blocks, as illustrated in Eq. (3):

(3) Y=OSR(X).

Secondly, the initial input X is subjected to a linear transformation to produce the query matrix Q for the ensuing attention computation, as seen in Eq. (4):

(4) Q=Linear(X).

Thirdly, the processed data Y undergoes a local refinement module LR (realised by a 3 × 3 deep convolution) and linear transformation, subsequently dividing into two components: the key K and the value V. Local refinement refers to the augmentation of feature details through deep convolution, as expressed in Eq. (5):

(5) K,V=Split(Linear(Y+LR(Y))).

Finally, the model encodes spatial relationships by calculating the dot product of Q and K, normalising by dividing by the square root of the number of channels ( d), adding a relative position bias matrix B to improve sensitivity to relative positions in the inputs, and subsequently applying the Softmax function to produce the attention matrix. This matrix is multiplied by V to yield the final output Z. The entire procedure is illustrated in Eq. (6):

(6) Z=Softmax(QKTd+B)V.

The incorporation of the OSRA module into the backbone network facilitates enhanced feature processing and information extraction tailored to the specific needs of coal-rock image recognition. The OSRA module’s sophisticated overlapping design markedly improves the network’s capacity to analyse high-resolution input images, particularly in preserving and representing information at the image’s edge areas. In addition, the implementation of OSRA boosts spatial fidelity of features and improves the model’s capacity to integrate local information with global context through the refinement of the attention mechanism. In summary, the technique significantly enhances the precision of coal-rock image recognition technology and exemplifies the successful integration of theoretical innovation with practical application.

Slim-neck paradigm

This study introduces the slim-neck paradigm, comprising the grouped shuffle convolution (GSConv) module and the variety of view-grouped shuffle cross stage partial (VoV-GSCSP) module, to improve the model’s lightness without sacrificing accuracy.

GSConv is an efficient convolution technique that integrates standard convolution (SC), depth-wise separable convolution (DSC), and shuffle operations, enhancing the feature representation capacity and computational efficiency of convolutional networks. The configuration is depicted in Fig. 6. In this strategy, the standard convolution thoroughly processes the input feature map to guarantee information completeness; however, it incurs significant computational costs. DSC utilizes a stepwise processing approach, wherein the characteristics of each input channel are initially processed independently via channel-by-channel convolution, followed by aggregation of these features using 1 × 1 convolution. This method effectively diminishes the number of references and FLOPs, yet it may result in the segregation of information among channels. GSConv addresses this issue by implementing the shuffle operation to uniformly recombine the outputs of each channel following DSC, enabling each output channel to incorporate information from various input channels. This approach enhances the network’s nonlinear expressive capacity and feature integration capability without imposing extra computational demands (Zhang et al., 2025; Dong et al., 2025).

Figure 6 The structure of GSConv module.

‘SC’ stands for standard convolution and ‘DSC’ stands for depth separable convolution, each of which contains a batch normalisation layer and an activation layer.

Assuming W and H represent the width and height of the input feature map, respectively, while C1 and C2 signify the number of input and output channels, with K1×K2 representing the dimensions of the convolutional kernel. The computational expense of the standard convolution is delineated in Eq. (7), wherein all input and output channels are entirely interconnected, and each element of the convolution kernel necessitates a multiply-add operation with the corresponding element of the input feature map.

(7) CostSC=W×H×C1×K1×K2×C2.

Equation (8) illustrates the computational cost of the DSC, which markedly diminishes the number of parameters and computations by initially performing spatial convolution on each input channel independently, followed by the aggregation of these results using a 1×1 convolution.

(8) CostDSC=W×H×C1×K1×K2+W×H×1×1×C1×C2.

Therefore, the computed cost ratio of GSConv to SC is presented in Eq. (9):

(9) ratioc=12⋅W×H×K1×K2×C2×(C1+1)W×H×C1×K1×K2×C2=C1+12C1→12.

That is, the computational expense of GSConv is approximately 50% of that of standard convolution, yet its contribution to the model’s learning capacity is analogous to that of standard convolution.

Moreover, GSConv is especially appropriate for the model’s neck region, characterized by high feature concentration and dimensional compression. The implementation of GSConv minimizes information redundancy, thereby significantly enhancing the model’s efficiency and performance. This design optimizes computational resource utilization and enhances the model’s accuracy in processing complex input data, resulting in superior performance while maintaining lightweight, making it especially appropriate for deployment on resource-constrained edge devices.

VoV-GSCSP is a sophisticated convolutional network module aimed at enhancing the computational efficiency and learning capacity of the network. It integrates the cross stage partial network (CSP) module with the GS bottleneck module based on the GSConv module to facilitate efficient information flow and feature reutilization via a distinctive channel segmentation and reorganization strategy. This design minimizes inter-layer dependencies and lowers memory demands for forward and backpropagation, while enhancing the speed and precision of model inference. Through the implementation of advanced channel management policies, VoV-GSCSP can markedly decrease computational complexity while maintaining network performance. Furthermore, the module has exhibited significant hardware compatibility in practical applications, rendering it especially advantageous in resource-limited settings. Figure 7 illustrates the configuration of the GS bottleneck and VoV-GSCSP.

Figure 7 (A and B) GS Bottleneck module and VoV-GSCSP module architecture.

Slim-neck is a design paradigm for object detection modeling, comprising flexible GSConv modules and VoV-GSCSP modules, intended to offer a streamlined and efficient network architecture for object detection on edge devices. This study demonstrates that slim-neck enhances model accuracy and feature map processing efficiency by substituting the standard convolutional layer with a GSConv module at the network’s neck and replacing the traditional C2f module with a VoV-GSCSP module. This design significantly diminishes computational complexity and memory demands, positioning the slim-neck architecture as an exceptional solution for real-time detection tasks, particularly in settings with constrained computational and storage resources.

Experimental setup

This section details the construction of the coal-rock dataset, outlines the configuration of the experimental environment and model parameters, and presents the evaluation metrics employed in the experiment.

Dataset description

This study utilised a dataset supplied by the School of Mining Engineering at Heilongjiang University of Science and Technology (Wang, Zhao & Xue, 2024), comprising 70 high-resolution photographs sourced from two categories of underground coal mines. The images were obtained using Olympus TG-320 and Huawei PCT-AL10 cameras, featuring resolutions of 4,288 × 3,216 and 4,000 × 3,000 pixels, respectively, as illustrated in Fig. 8.

Figure 8 (A and B) Images from different coal mining scenarios.

Image credit: Wang et al. (2024). Coal and Rock [Data set]. Zenodo. https://doi.org/10.5281/zenodo.10702704.

Step 1: Image capture & preprocessing. Initially, all images were consistently resized to 4,000 × 3,000 pixels using a bicubic interpolation algorithm to ensure uniform processing. Each image was then divided into 12 non-overlapping blocks of 1,000 × 1,000 pixels through uniform segmentation, achieved by cropping four times horizontally and three times vertically. Out of the 840 blocks obtained through segmentation, 527 blocks were retained based on the inclusion of distinctly identifiable coal mine targets, which were required to meet a minimum target size of 50 × 50 pixels and a minimum contrast level of 10% difference. These blocks were subsequently manually labelled using the LabelImg tool, resulting in a total of 813 coal targets being identified. The labelling process specifically excluded poorly illuminated or out-of-focus areas and focused exclusively on a single category: coal. This approach was taken to ensure data quality and the precision of model training.

Step 2: Data splitting. The preprocessed images are randomly allocated into training, validation, and test sets in the proportions of 70%, 20%, and 10%, containing 368, 106, and 53 images, respectively. The training set is primarily used to provide a sufficient volume of data, enabling the model to learn the intricate aspects of coal imagery and thereby achieve robust generalization capabilities. The validation set is employed for ongoing performance evaluation during model training, facilitating parameter adjustments to ensure model stability and reliability. The test set serves as an independent dataset to evaluate the final performance of the model and predict its effectiveness in real-world applications. This allocation strategy provides a solid foundation for model development and evaluation, ensuring the reliability and validity of the experimental results.

Step 3: Data enhancement. To enhance data diversity and simulate various shooting perspectives that might be encountered in practical scenarios, offline data augmentation was performed, primarily utilizing image panning techniques. Starting from the upper left corner of each 1,000 × 1,000 pixel block, multiple crops are made at 180-pixel intervals in both horizontal and vertical directions. Each block is processed to produce nine overlapping images, each with dimensions of 640 × 640 pixels. This augmentation technique aims to improve the model’s ability to recognize features across different positions and size variations.

Following the aforementioned processing and enhancement techniques, the dataset now includes 4,743 images, with 3,312 allocated to the training set, 954 to the validation set, and 477 to the test set. The processing workflow of the dataset is illustrated in Fig. 9. This comprehensive methodology for processing and labeling ensures the high quality and applicability of the dataset, providing a solid foundation for subsequent model training and testing. The original dataset is available at Zenodo: https://doi.org/10.5281/zenodo.10702704. The processed dataset is available at Zenodo: https://doi.org/10.5281/zenodo.10702879.

Figure 9 Dataset processing flow.

Image credit: Wang et al. (2024). Coal and Rock [Data set]. Zenodo. https://doi.org/10.5281/zenodo.10702704.

Setup and configuration

The model training and evaluation for this study were conducted on a high-performance server equipped with the following specifications: an Intel(R) Xeon(R) Platinum 8,358 P processor running at 2.60 GHz, 80 GB of system memory, and an NVIDIA GeForce RTX 3,090 graphics card with 24 GB of video memory. The system runs on Ubuntu 20.04 LTS and is equipped with CUDA 11.8 and CuDNN 8.4.0. The programming language used is Python 3.8, the deep learning framework is PyTorch 2.0.0, and the image processing library is torchvision 0.15.2.

The experimental setup utilised the parameter configurations displayed in Table 1. Additionally, the default values of the YOLOv8n model were employed for all remaining training parameters.

Table 1 Model parameter configuration for YOLOv8-POS.

Parameter	Explanation	Value	
Weights	Pre-training weights file	YOLOv8n.pt	
Epochs	Training epochs	300	
Patience	Early stop mechanism epochs	50	
Batch	Batch size	32	
imgsz	Image size	640	
Workers	Number of data loading threads	15	
Optimizer	Optimizer	SGD	
lr0	Initial learning rate	0.01	
lrf	Final learning rate	0.937	
Momentum	SGD momentum	0.0005	
Weight_decay	Optimizer weight decay	YOLOv8n.pt	

Evaluation metrics

This study assessed the model’s performance using many measures, including accuracy, recall, average precision (AP50, AP50:95), number of parameters (Params), and number of floating point operations (FLOPs).

Accuracy is the measure of the likelihood of correctly identifying a positive class out of all the samples anticipated to be a positive class. It is defined by Eq. (10):

(10) Precision=TPTP+FP

where TP represents the count of positive cases that were accurately classified, while FP represents the count of negative cases that were inaccurately labelled.

Recall, as stated in Eq. (11), is the likelihood of correctly predicting a positive class in a sample that is truly a positive class:

(11) Recall=TPTP+FN

where FN represents the count of positive examples that were classified incorrectly.

This study specifically examines the recognition of targets belonging to a particular category in coal-rock images. As a result, it opts to use Average Precision (AP) instead of mean Average Precision (mAP) to assess the model’s detection capabilities. Average Precision (AP) is employed to evaluate the model’s performance on particular categories and offers an intuitive performance metric. The calculation formula is expressed by Eq. (12):

(12) AP=∫01P(R)dR

where P(R) represents the precision when a specific recall value R is considered. AP50 is the mean precision when the intersection over union (IoU) threshold is set to 0.5. AP50:95 represents the average of the mean precision at various IoU thresholds, ranging from 0.50 to 0.95 in increments of 0.05, resulting in a total of 11 thresholds.

The number of parameters (Params) refers to the overall number of parameters that need to be trained throughout the model training process. This count indicates the spatial complexity of the model. The number of FLOPs denotes the quantity of floating-point operations necessary for the model to execute a forward propagation process, which represents the time complexity of the model. A model with fewer parameters and FLOPs is considered lighter, making it suited for application settings with limited resources.

Results and discussion

This section describes in detail the ablation experiments and the comparison experiments, and demonstrates the visual analysis of the experimental results in order to fully evaluate the improved YOLOv8-POS model.

Ablation experiments

This study conducted eight sets of ablation experiments to evaluate the influence of various components on model performance, utilizing the same dataset and parameter configuration for each set. Experiment 1 employed the original YOLOv8n network as a baseline model without including any enhancements. The subsequent tests (Experiment 2 to Experiment 4) incorporated three distinct components into the baseline model: the C2f-PConv, the OSRA, and the Slim-neck, respectively. In Experiments 5 to 7, two-by-two combinations of these three components were evaluated, with Experiment 5 integrating C2f-PConv and OSRA. Experiment 6 integrates C2f-PConv with slim-neck, whereas Experiment 7 amalgamates OSRA with Slim-neck. Ultimately, Experiment 8 amalgamates all three components to establish the YOLOv8-POS network presented in this research. This configuration seeks to clearly assess the influence of each distinct component and their combinations on every performance parameter. Table 2 presents the detailed experimental results, whereas Fig. 10 illustrates the various performance metrics of YOLOv8-POS during training and validation.

Table 2 Results of ablation experiments.

Bold entries represent the highest values for each metric.

No.	C2f-PConv	OSRA	Slim-neck	Precision
(%)	Recall
(%)	AP50
(%)	AP50:95
(%)	Params
(M)	FLOPs
(G)	
1	×	×	×	90.7	68.2	74.7	59.5	3.01	8.2	
2	✓	×	×	90.5	67.6	75.9	60.5	2.61	7.0	
3	×	✓	×	88.9	69.7	76.5	61.5	3.21	8.4	
4	×	×	✓	90.8	67.9	75.4	61.7	2.80	7.4	
5	✓	✓	×	87.8	71.1	76.6	61.9	2.81	7.2	
6	✓	×	✓	89.3	69.1	76.2	62.0	2.40	6.2	
7	×	✓	✓	91.3	71.4	76.6	62.8	3.00	7.5	
8	✓	✓	✓	90.4	72.4	77.1	63.6	2.60	6.4	

Figure 10 Performance indicators of YOLOv8n-POS.

The findings indicate that the C2f-PConv module markedly diminishes model complexity by refining the convolutional architecture and enhancing the network’s computational efficiency, yielding improvements of 1.2% and 1.0% for AP50 and AP50:95, respectively, alongside a decrease in parameters and FLOPs by 0.40 M and 1.2 G, respectively, thereby alleviating the model’s burden while preserving a degree of performance enhancement. The OSRA module allows the model to dynamically concentrate on essential visual features, enhancing its sensitivity to critical information, resulting in improvements of 1.8% and 2.0% for AP50 and AP50:95, respectively. Despite an increase of around 0.20 M parameters and 0.2G FLOPs, the substantial performance enhancements demonstrated the cost-effectiveness of the enhanced resource allocation, underscoring OSRA’s efficacy in augmenting the model’s sensitivity to the target. The slim-neck paradigm diminishes the model’s computing demands for visual tasks by streamlining the network’s ‘neck’ structure, achieving improvements of 0.7% in AP50 and 2.2% in AP50:95, while concurrently decreasing the number of parameters and FLOPs by 0.21 M and 0.8 G, respectively. It significantly enhances model performance at elevated IoU thresholds while decreasing both the parameter count and computational demands of the model.

When the three components, C2f-PConv, OSRA, and slim-neck, are concurrently incorporated into the YOLOv8n model, the model attains peak performance regarding the primary performance metrics, demonstrating enhancements of 2.4% and 4.1% for AP50 and AP50:95, respectively, in comparison to the baseline model. The parameters and FLOPs have been greatly optimized, with reductions of 0.41 M and 1.8 G, respectively. This demonstrates that the three components exert a substantial synergistic effect on enhancing detection accuracy and model efficiency, rendering the model both efficient and energy-conserving, hence validating the efficacy of the new methodology.

Comparison experiments

Coal-rock recognition activities are essential for mining automation, necessitating high model accuracy, real-time performance, and efficiency. This study conducted a comparative analysis of various advanced detection models, including Faster R-CNN (Ren et al., 2016), YOLOv3 (Redmon & Farhadi, 2018), YOLOv5n, YOLOv5s, YOLOv7 (Wang, Bochkovskiy & Liao, 2022), YOLOv8n, YOLOv10n, YOLOv10s (Wang et al., 2024), and Gold-YOLOn (Wang et al., 2023), to accurately assess the performance of the YOLOv8n-POS model in coal-rock image recognition tasks. The dataset and experimental parameters were consistent across all experimental groups. These experiments seek to demonstrate the benefits of YOLOv8n-POS regarding coal-rock recognition accuracy and computing efficiency, as well as its applicability in particular mining contexts. The experimental findings are shown in Table 3, and the Precision-Recall curves for several algorithms are illustrated in Fig. 11.

Table 3 Results of comparison experiments.

Bold entries represent the highest values for each metric.

Model	Precision
(%)	Recall
(%)	AP50
(%)	AP50:95
(%)	Params
(M)	FLOPs
(G)	
Faster R-CNN	56.5	75.3	75.0	51.6	137.10	370.2	
YOLOv3	88.7	65.4	73.2	57.0	61.52	155.3	
YOLOv5n	82.4	61.0	69.0	51.5	1.77	4.2	
YOLOv5s	87.9	62.0	70.4	55.0	7.02	15.9	
YOLOv7	91.2	66.0	73.8	57.1	37.20	105.1	
YOLOv8n	90.7	68.2	74.7	59.5	3.01	8.2	
YOLOv10n	89.4	68.7	75.2	57.7	2.30	6.7	
YOLOv10s	88.2	67.2	73.8	57.7	7.25	21.6	
Gold-YOLOn	94.0	67.7	73.6	59.2	5.60	12.1	
YOLOv8n-POS (Ours)	90.4	72.4	77.1	63.6	2.60	6.4	

Figure 11 (A–J) Precision-recall curves for different algorithms.

The YOLOv8n served as the baseline model and exhibited outstanding performance across multiple important performance parameters. The model attains an AP50 of 74.7% and an AP50:95 of 59.5%, surpassing the majority of the models evaluated. This performance results from both its efficient network architecture and its optimised feature extraction and processing techniques. Despite Faster R-CNN achieving somewhat superior performance on AP50, its substantial parameter count (137.10 M) and computational load (370.2GFLOPs) restrict its utility in resource-limited settings, particularly in mining locations where real-time efficiency is critical.

This work implements YOLOv8n-POS, derived from YOLOv8n, by incorporating enhanced components to improve overall model performance, resulting in an AP50 of 77.1% and an AP50:95 of 63.6%. Concurrently, the model’s parameter count is diminished to 2.60 M, and the computational load is decreased to 6.4GFLOPs, illustrating an exemplary equilibrium between performance and efficiency.

Incidentally, despite YOLOv10s utilising the novel compact inverted block CSP Bottleneck with two convolutions (C2fCIB) module, it underperforms relative to the significantly smaller YOLOv10n in certain coal-rock image recognition tasks. The C2fCIB module, while improved by deep feature iteration, boosts feature characterisation capacity but may result in overfitting due to heightened model complexity, hence impacting the model’s overall recognition accuracy.

In summary, YOLOv8n-POS not only retains the exceptional performance of YOLOv8n but also markedly enhances recognition accuracy and resource efficiency by structural advancements. The model’s performance in coal-rock image recognition unequivocally demonstrates its applicability and value, aligning with the study’s objective of creating an energy-efficient, high-performance recognition model, thereby substantiating the necessity and benefits of the enhanced method.

Visualization of experimental results

This study utilises YOLOv8n and its enhanced version YOLOv8n-POS to conduct a comprehensive analysis of the detection performance of coal and rock images in real-world application settings. The specific outcomes are presented in Fig. 12. The YOLOv8n-POS model in Group A effectively mitigates the problem of misclassifying debris and rock as coals. Within Group B, the model demonstrates enhanced precision in recognizing coals even when workers are obstructing the view, resulting in a significant decrease in the rate of erroneous detections. Within Group C, the enhanced model has the capability to recognize a greater number of fully-formed coals. The enhanced model in Group D and E adheres to the data annotation principles outlined in this work. It disregards challenging areas of the image that are affected by low light or blurring, avoids the acquisition of unstable features by the model, and enhances the model’s reliability. The complete study findings demonstrate that the YOLOv8n-POS model enhances both the recognition accuracy and the precision of coal target localisation to a significant extent.

Figure 12 Comparison of detection results.

Conclusions

This article proposes a YOLOv8-based model for coal-rock image recognition, termed YOLOv8-POS, to overcome the limitations of traditional YOLOv8 in coal-rock detection. The C2f-PConv module is introduced, which markedly diminishes computational resource consumption by selectively processing essential feature channels, thereby ensuring efficient operation in resource-constrained environments and proving more practical than conventional methods. The OSRA module integrated into the backbone network enhances the recognition of uneven illumination and edge areas, thereby significantly improving the model’s robustness and reliability in complex and variable downhole environments. Meanwhile, the slim-neck paradigm streamlines information flow and feature fusion, decreasing the computing burden and storage requirements of the model, making YOLOv8-POS more appropriate for real-time deployment on edge devices. The experimental results indicate that YOLOv8-POS demonstrates strong performance on the dataset utilized in this study, particularly in low-light conditions, while also accurately recognizing the distribution of coal and rocks, thereby confirming its applicability in practical scenarios. Compared to existing methods, YOLOv8-POS offers significant improvements in recognition accuracy and operational efficiency. While not yet implemented in a real-world setting, its modular design and lightweight structure consider the computational resource constraints of edge devices, theoretically enabling efficient operation under challenging conditions and offering a technical foundation for future deployment.

However, there are still limitations to this study. First, the dataset’s insufficient diversity, mostly focused on certain coal mining scenarios, may restrict the model’s applicability to different geological environments and illumination circumstances. Future research should broaden the dataset to encompass a wider range of geological contexts and diverse light conditions to enhance the model’s generalization capability. Second, the applicability of YOLOv8-POS to other mineral resource recognition remains unverified. Future research will investigate the model’s performance across diverse application scenarios through enhanced training and optimization, aiming to broaden its applicability in various fields. Furthermore, future research will aim to optimize data acquisition, image preprocessing, data enhancement, and model structure to minimize detection errors. The anticipated improvements are expected to broaden the application scope of YOLOv8-POS and enhance technical support for mineral resources exploration.

Supplemental Information

Supplemental Information 1 Literature table for coal-rock image recognition.

Some parts of this manuscript were edited with the assistance of ChatGPT. The authors take full responsibility for the accuracy and integrity of the content.

Additional Information and Declarations

Competing Interests

The authors declare that they have no competing interests.

Author Contributions

Yanqin Zhao conceived and designed the experiments, performed the experiments, analyzed the data, performed the computation work, authored or reviewed drafts of the article, and approved the final draft.

Wenyu Wang performed the experiments, analyzed the data, performed the computation work, prepared figures and/or tables, and approved the final draft.

Data Availability

The following information was supplied regarding data availability:

The code of YOLOv8-POS is available at Zenodo: Jason_2k. (2024). Jason-2k/YOLOv8-POS: V 1.0 (V1.0). Zenodo. https://doi.org/10.5281/zenodo.13777691.

The model weights are available at Zenodo: Wang, W. (2024). The model weights file of YOLOv8-POS. Zenodo. https://doi.org/10.5281/zenodo.13777703.

The original dataset is available at Zenodo: Wang, W. (2024). Coal and Rock [Data set]. Zenodo. https://doi.org/10.5281/zenodo.10702704.

The processed dataset is available at Zenodo: Wang, W. (2024). Coal and Rock after processing in YOLO format [Data set]. Zenodo. https://doi.org/10.5281/zenodo.10702879.

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
