# Peer review of "YOLOv8-POS: a lightweight model for coal-rock image recognition"

_PeerJ Computer Science, doi:10.7717/peerj-cs.2820_

## Round 0.1 · original submission · Major Revisions

The reviewers acknowledge the significance of the topic explored in the paper and agree on the validity of the proposed approach. However, revisions are necessary, particularly concerning the presentation, comparison with other methods, and clarification of certain design choices. Please refer to the reviews for further details.

Reviewer 1 ·

Basic reporting

The background and introduction give perspective, and the content is pertinent and would enhance if added a few well-referenced and related journals. To improve the reader's comprehension of the general strategy, the paper does not include a clear representation of the flow diagram for the suggested methodology.

The C2f structure's various feature extraction layers will raise memory and computational cost, especially for the intricate coal rock inputs. How do you overcome it?

The motivation for the study is not stated properly in the introduction, nor is the subject sufficiently introduced. A more thorough description of the study environment, its importance, and the gaps it seeks to fill would improve the manuscript's foundation.

The manuscript's formal results must ensure clarity by offering accurate conclusions for each of the suggested logical constructions. Readers will gain a better understanding of the work's underlying principles and ramifications as a result.

Experimental design

The paper's content fits within the specified article category and is in good alignment with the journal's goals and scope.

The current coal-rock dataset requires significant preprocessing or data augmentation to address real-world challenges such as varying illumination, low light, and low contrast conditions. Could the authors clarify the measures taken to simulate or mitigate these issues during data preparation and model training to match the real-time scenario?

Using OSRA, the Overlapping regions will lead to redundant information processing, which could reduce the overall efficiency of feature extraction. Can you justify with proper validations why only Overlapping Spatial Reduction Attention would fit the job?

Model designs employ the Slim-neck paradigm to minimize the amount of parameters between the head and backbone. However, the accuracy of the model may be impacted by excessive neck constriction, which may discard significant spatial and semantic data.

The integration of YOLOv8-POS, C2f-PConv module, Overlapping Spatial Reduction Attention module, and Slim-neck design must be thoroughly explained in order to fully comprehend the suggested model. It is also possible to justify the need for this combination.

Can the suggested model be used to different datasets and challenges, or is it limited to the particular problem scenario?

To assist the work, I would recommend adding a few recent work references to each of the model's suggested components.

Validity of the findings

The conclusions are well articulated. However, they are primarily limited to summarizing the supporting results. Furthermore, the findings presented can be improved to adequately showcase the model's accomplishments and potential.

Indeed, the tests and assessments carried out are adequate and show the effectiveness of the suggested model.

Additional comments

The work is of good quality and demonstrates significant contributions. It requires only minor revisions before being recommended for acceptance.

Cite this review as

·

Basic reporting

'no comment'

Experimental design

'no comment'

Validity of the findings

'no comment'

Additional comments

The authors' article is devoted to an important and relevant issue related to the development of a model for image recognition of coal and rocks.
This paper presents a new approach to image recognition of coal and rocks, called YOLOv8-POS, aimed at reducing the number of false positives and increasing accuracy while reducing computational complexity. Common problems such as blurred images, insufficient lighting and cluttered scenes make it much more difficult for existing models to work effectively. YOLOv8-POS solves these problems by using innovative architectural solutions. The key element of YOLOv8-POS is the C2f-PConv module, which effectively combines the advantages of C2f and partial convolution. This module allows you to selectively process image channels, minimizing unnecessary calculations and preserving critical information about the target objects. Additional improvement is achieved due to the built-in spatial reduction control module, which optimizes the integration of spatial objects and increases the stability of the model to complex scenarios. The use of a bottleneck design further reduces computational requirements and memory requirements. The experimental results demonstrate the significant superiority of YOLOv8-POS over existing methods. The model reaches AP50 in 77.1% and AP50:95 in 63.6% with a significant reduction in the number of parameters (up to 2.60 million) and computational complexity. This is confirmed by a comparative analysis with other well-known algorithms. Thus, YOLOv8-POS is an effective and practically applicable approach to solving problems of image recognition of coal and rocks in difficult conditions.
In conclusion, it should be noted that further research will be aimed at adapting YOLOv8-POS to work with video streams, which will allow it to be used in real time to monitor mining operations. It is also important that it is planned to explore the possibility of expanding the functionality of the model to recognize a wider range of objects and materials in the mining industry.

However, it would be necessary to clarify a number of comments that are available to the article:
1. The introductory part of the article talks about the creation of an improved YOLOv8-POS model, but the results of comparison with the basic YOLOv8 model are not presented. This is important for evaluating the effectiveness of the changes made. Without such a comparison, it is difficult to assess the real improvement in accuracy and speed.
2. In the paper, it should be explained on which data set the model was trained? What is the size of the dataset, its composition, and the class distribution? Information about the dataset is necessary to understand the generalizing ability of the model.
3. The section "Traditional methods and machine learning" lists various methods and their accuracy, but does not conduct a systematic comparative analysis. What are the advantages and disadvantages of each method? What factors affect the accuracy of the classification? There is no table with generalized results for convenient comparison.
4. The paper should focus more clearly on machine learning methods based on the use of neural networks to solve the tasks and also point out greater reliability of author's method compared to deterministic methods.
5. It is not entirely clear whether YOLOv8 has been compared with other well-known object detection models (for example, Faster R-CNN, SSD, EfficientDet). Such a comparison is necessary to assess YOLOv8's competitive advantages. By what criteria (accuracy, speed, model size) is YOLOv8 superior to other models?
6. In the section "YOLOv8 algorithm overview" we are talking about the improved architecture of the local attention management network (ELAN) in the C2f module. What is this improvement? What changes have been made to the ELAN architecture? Without a more detailed description, it is difficult to assess the contribution of these changes to improving efficiency.
7. Should we elaborate in more detail on the characteristics of the equipment with which the experiments were carried out? What libraries and frameworks were used? A detailed description of the experimental conditions is necessary for reproducibility of the results.
8. How was the study of the errors made by the model conducted? Error analysis can help identify the weaknesses of the model and guide further research.
9. The paper considers 70 source images. This is a very small dataset for training a complex deep learning model such as YOLOv8. Even after zooming in to 4,743 images by cropping, the question of data sufficiency remains open. A small data set can lead to overfitting of the model and poor generalizing ability. It is necessary to specify the number of objects of each class in the dataset and justify the sufficiency of this number for training the model.
10. The text of the article indicates that 527 of the 840 blocks were selected, "using a criterion that required the presence of clearly identifiable coal mining facilities." This criterion is not quantified. How was "clear identification" defined? Subjective selection can lead to a systematic error in the dataset. It is necessary to describe the block selection criteria more precisely and, preferably, provide quantitative indicators (for example, the minimum object size, the minimum contrast level).
11. In conclusion, it is necessary to clarify the parameters of the C2f-PConv and OSRA modules, as well as the "bottleneck" paradigm, with a detailed description of their implementation and functionality. How exactly were these modules integrated into the YOLOv8 architecture? What parameters have been changed? How do these modifications affect the image processing process? Without this description, it is impossible to evaluate the scientific novelty and contribution of the work.
Best regards.

Cite this review as

·

Basic reporting

1. Introduction and Novelty

Expand the introduction to clearly articulate the broader applicability of YOLOv8-POS beyond coal mining, emphasizing its relevance to other fields such as geology, construction, and agriculture.
Clearly state the limitations of previous models, particularly YOLOv8, to establish the novelty of the proposed enhancements. Highlight how the combination of C2f-PConv, OSRA, and Slim-neck modules introduces innovation in lightweight object detection.
Strengthen the discussion on how YOLOv8-POS addresses specific challenges like image defocus, dim lighting, and occlusions, which are critical in underground environments.

2. Dataset and Generalization

Provide more details about the diversity of the dataset, particularly variations in lighting, geological conditions, and coal quality. Explain how this diversity impacts the robustness of the model.
Discuss potential limitations due to the dataset's focus on specific coal mining scenarios and suggest strategies to expand its scope for better generalization.
Address the potential for applying the model to other mineral recognition tasks or industrial applications.

3. Methodology and Experimental Design

Provide additional justification for the selection of C2f-PConv, OSRA, and Slim-neck components. Explain the trade-offs between accuracy and computational efficiency.
Include specific parameter settings for the ablation experiments to ensure reproducibility.
Discuss the challenges in deploying YOLOv8-POS in real-world resource-constrained environments and how the model addresses them effectively.

4. Application and Practical Utility

Emphasize the practical advantages of YOLOv8-POS in underground mining operations, such as real-time hazard detection and resource monitoring on limited-resource edge devices.
Discuss the potential for scalability to other domains, such as defect detection in industrial settings or analyzing geological formations.
Highlight any real-world testing conducted, and if not, suggest its inclusion to validate the model's practical applicability.

5. Figures and Visualizations

Ensure all figures, such as Figures 3 and 5, are accompanied by clear legends and detailed explanations in the text to improve reader comprehension.
Add a comparative visualization of YOLOv8-POS’s performance against other lightweight detection frameworks to emphasize its superiority.

6. Writing and Presentation

Improve the readability of technical descriptions, especially in the abstract and introduction. Simplify overly complex sentences while maintaining technical precision.
Address minor grammatical issues and ensure the language is polished for international audiences.
Reorganize key sections to maintain a logical flow, particularly when discussing experimental results and their implications.

7. Results and Discussion

Strengthen the conclusion by summarizing the key contributions and practical implications of YOLOv8-POS.
Discuss future directions, such as testing the model in diverse geological settings or adapting it for other mineral resources.
Include specific recommendations for addressing limitations, such as improving generalization and increasing dataset diversity.

8. Novelty and Comparative Advantage

Highlight the distinctiveness of YOLOv8-POS compared to other state-of-the-art models, particularly in terms of lightweight design and computational efficiency.
Discuss the novelty of integrating C2f-PConv, OSRA, and Slim-neck into YOLOv8 architecture for challenging recognition tasks.
Emphasize the model's potential for broader impact, such as in environmental monitoring or industrial quality assurance.

Experimental design

none

Validity of the findings

none

Additional comments

improve the literature further.
1. Using the characteristics of infrared radiation during the process of strain energy evolution in saturated rock as a precursor for violent failure
2. Prediction of an early failure point using infrared radiation characteristics and energy evolution for sandstone with different water contents
3. Application of machine learning and multivariate statistics to predict uniaxial compressive strength and static Young’s modulus using physical properties under different temeperture
4. Prediction of sandstone dilatancy point in different water contents using infrared radiation characteristic: Experimental and machine learning approaches
5. Early violent failure precursor prediction based on infrared radiation characteristics for coal specimens under different loading rates

Cite this review as

---

## Round 0.2 · accepted · Accept

The reviewers' concerns have been addressed and the paper has been improved greatly with respect to the previous version. Hence, I believe it is ready for publication.

·

Basic reporting

The article conform to professional standards of courtesy and expression

Experimental design

The investigation have been conducted rigorously and to a high technical standard.

Validity of the findings

Conclusions are well stated & limited to supporting results.

Cite this review as